# Large Point-to-Gaussian Model for Image-to-3D Generation

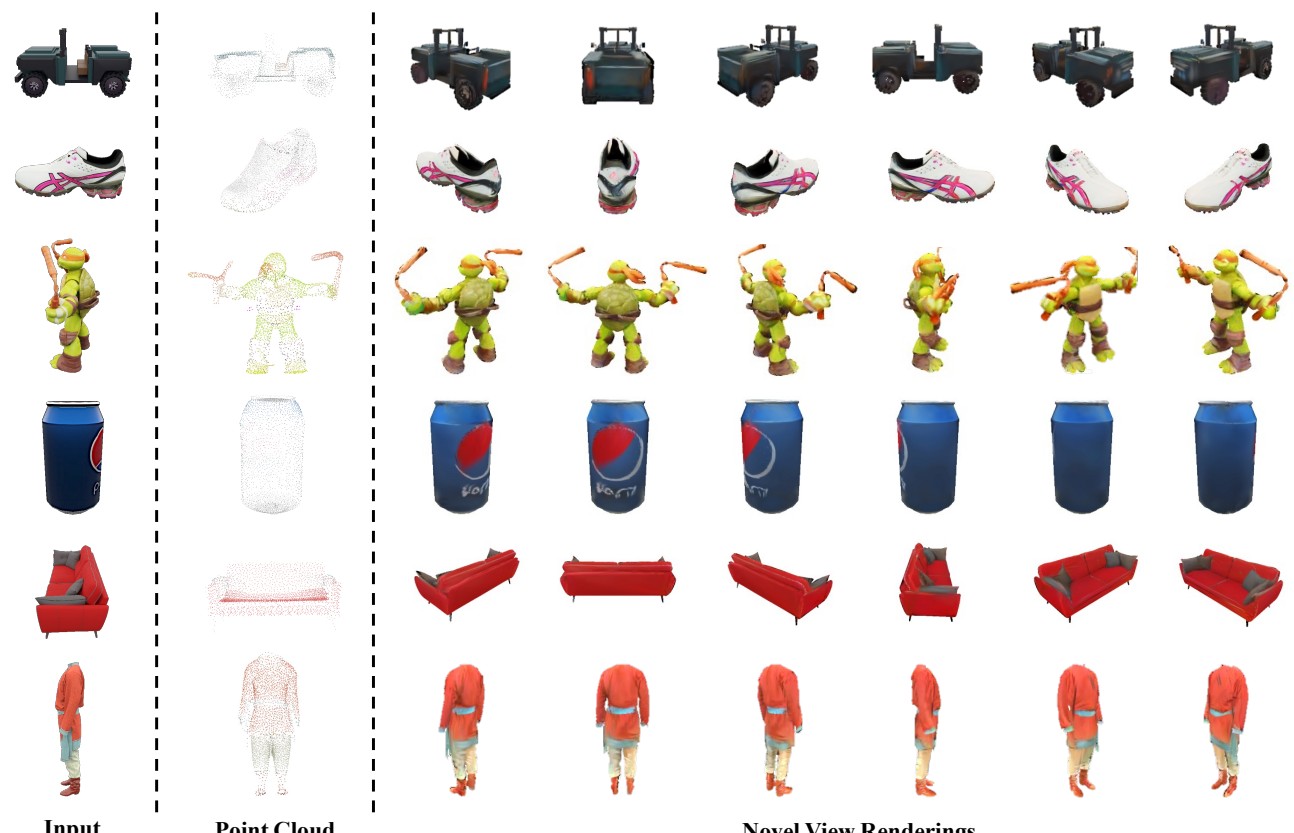

**Input**  **Point Cloud**  **Novel View Renderings**

**Figure 1: Our method produces high-fidelity generation results from a single-view image.**

## ABSTRACT

Recently, image-to-3D approaches have significantly advanced the generation quality and speed of 3D assets based on large reconstruction models, particularly 3D Gaussian reconstruction models. Existing large 3D Gaussian models directly map 2D image to 3D Gaussian parameters, while regressing 2D image to 3D Gaussian representations is challenging without 3D priors. In this paper, we propose a large Point-to-Gaussian model, that inputs the initial point cloud produced from large 3D diffusion model conditional on 2D image to generate the Gaussian parameters, for image-to-3D generation. The point cloud provides initial 3D geometry prior for Gaussian generation, thus significantly facilitating image-to-3D Generation. Moreover, we present the **A**ttention mechanism, **P**rojection mechanism, and **P**oint feature extractor, dubbed as **APP** block, for fusing the image features with point cloud features. The qualitative and quantitative experiments extensively demonstrate the effectiveness of the proposed approach on GSO and Objaverse datasets, and show the proposed method achieves state-of-the-art performance.

## CCS CONCEPTS

• **Information systems** → **Multimedia content creation**; • **Computing methodologies** → **Appearance and texture representations**; **Virtual reality**.

## KEYWORDS

3D Generation, 3D Gaussian Splatting, Single-View Reconstruction, Point Cloud

# 1 INTRODUCTION

Generating high-quality 3D assets from images is a pivotal task in numerous fields, notably in gaming, film production, and VR/AR, etc. The learning-based 3D generation algorithms [12, 33] allow rapid generation of high-quality 3D assets free of tedious manual processes and complex computer graphics tools.

Recently, the realm of 3D generation has witnessed a surge of innovative techniques, with particular prominence given to two main approaches: 2D-lifting-based generation and feed-forward generation, driving the field's progress. Following the pioneering work [33], the 2D-lifting approaches [17, 33, 50] leverages Score Distillation Sampling (SDS) [33] to distill 3D implicit representations (e.g., NeRF [30]) from large diffusion models [20, 21], while feed-forward generation approaches [12, 15, 45, 51] straightforward reconstruct 3D implicit representations from 2D image without iterative optimization. Compared to 2D-lifting approaches that require extensive optimization time, the feed-forward generation techniques can generate 3D assets within a few seconds. Besides, inspired by recent 3D Gaussian Splatting (3D-GS) [14] with promising rendering quality in the novel view synthesis (NVS) and fast rendering speed, the latest approaches improve the speed and quality of 3D generation by integrating the 3D-GS into 2D-lifting generation [6, 41, 52] or feed-forward generation [40, 49, 55].

The previous feed-forward approaches with 3D-GS [40, 49, 55] build mappings directly from implicit image features to Gaussian parameters. However, this regression based method is non-trivial for 3D-GS learning, given that the input 2D image does not always contain efficient 3D information for the corresponding object. By contrast, Point Cloud as an effective 3D representation is capable of providing informative geometry priors for the generation of explicit 3D Gaussians, whilst the current large 3D diffusion model (e.g. Point-E [31]) can generate diverse and satisfying initial point cloud for the input image. Therefore, we present to generate Gaussian parameters from point cloud, dubbed as Point-to-Gaussian, to advance the 3D-GS learning for image-to-3D generation as shown in Fig. 2.

Nevertheless, it is still non-trivial to convert the coarse point cloud from 3D large models to Gaussian representations given that the generated point cloud might sparse and noisy, which carrying insufficient appearance feature and inaccurate geometry structural information to provide informative prior for precise Gaussians generation. We thus first utilize a point cloud upsampler to densify it and thus enhance the features, and then utilize a point feature extractor to extract the feature from the point cloud. Besides, considering that the input image contains rich appearance information, we present to enhance the corss modality features to fuse the 2D image representations to the 3D point cloud for facilitating the 3D representations. Specifically, we project the 3D point to 2D image according to camera pose to obtain the features. However, there are occlusions in the point cloud during the projection. We introduce an attention mechanism to selectively query features for 3D points, enhancing their representations, particularly under occlusion. Subsequently, we integrate the 3D representations obtained from the **A**ttention mechanism, pose-aware **P**rojection mechanism, and **P**oint cloud feature extractor, termed APP Block, to conduct cross modality enhancement for more effective Gaussian learning.

The Gaussian parameters are finally generated using a multi-head Gaussian decoder, and the novel view images are rendered by conventional Gaussian splatting.

Our method converges quickly, which is trained only on the objaverse-LVIS [10] subset and achieves the comparable results of previous state-of-the-art methods trained with much more data, which indicates the effectiveness of the proposed method.

In summary, our main contributions are as follow:

- We propose a novel framework to generate high-quality 3D Gaussians with point cloud input. To our knowledge, our method is the first attempt to utilize the generalizable Point-to-Gaussian Generator for feed-forward image-to-3D generation.
- We introduce the **A**ttention mechanism, **P**rojection mechanism, and **P**oint feature extractor as APP block into Point-to-Gaussian generator for Cross Modality Enhancement, which further integrate the geometric structural features with the 2D texture features for more effective learning.
- Qualitative and quantitative results demonstrate that our method achieves comparable performance with the previous state-of-the-arts, even trained with much smaller dataset.

# 2 RELATED WORK

## 2.1 Diffusion Priors for 3D Generation

Recent efforts for generating 3D contents are mostly inspired from the success of 2D generative models [35]. They usually rely on Score Distillation Sampling (SDS) proposed in DreamFusion [33], which minimizes the difference between rendered images from the 3D object and the 2D diffusion with CLIP priors [34]. [43] interprets predictions from pretrained diffusion models as a score function of the data log-likelihood to optimize 3D representations via score matching. Subsequent work has made further enhancements based on SDS-based optimization. [17] increased the resolution of the generation, and introduced direct mesh optimizations. [47] proposed Variational Score Distillation (VSD) to enhance the quality and diversity of the generations. [50] introduced 3D shape prior from text-to-shape for better alignment between text and 3D shape. [4, 16, 36, 38] are proposed to achieve substantial improvement in quality, and further alleviate the Janus problem. [29, 41] aim to improve the optimization speed of 3D generation. Besides, many efforts [19–22, 24, 27, 37, 44] extend text-to-3D to single-view 3D generation. Theses methods focus on exploring non-optimization paradigms, and propose multi-view diffusion models which incorporates the information between different views for more consistent multi-view generation. Then the 3D reconstruction approaches are applied to obtain the 3D representations. SV3D and V3D [8, 42] are proposed to leverage video diffusion models for generating more consistent multi-view images.

## 2.2 Single-Stage 3D Generation

Unlike the SDS-based approach, the single-stage 3D generation method obtains 3D representation directly from image or text by a single feed-forward. Previous works attempt to train 3D diffusion models directly on point clouds or volumes [3, 13, 31]. However,

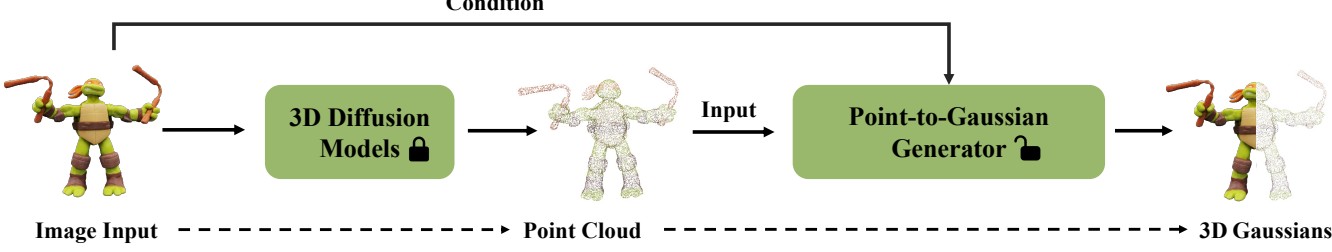

**Figure 2: Illustration of our full pipeline. Given a single-view image as input, the corresponding point cloud is first generated via pre-trained 3D diffusion model [13]. Then the point cloud is sent as the input of our proposed Point-to-Gaussian Generator. The input image is introduced as complementary condition to our Gaussian generator.**

they do not generalize well to other scenes and cannot provide satisfactory textures. Recently, LRM [12] first utilizes large transformers trained on large-scale 3D datasets [9, 10] to directly predict triplane NeRF from single view in a few seconds. [45] and [15] extend the input from single-view to sparse views. [15, 51] combine the large reconstruction model with diffusion priors to achieve text-to-3D and image-to-3D generation. However, these methods utilize volume rendering based triplane NeRF as the 3D representation, which still require massive forward inferences and computation.

### 2.3 Gaussian Splatting in 3D Generation

3D Gaussian Splatting (3D-GS) [14] showed marvelous performance in novel view synthesis for single scene optimization. This representation has been applied in many downstream tasks, such as generalizable reconstruction [1, 5, 39], human reconstruction [54], and 4D generation [18], etc. Some concurrent works [7, 41, 52] adopt 3D-GS for SDS-based optimization to decrease generation time. In the field of single-stage 3D generation, [55] combines LRM with 3D-GS for faster rendering speed and superior rendering quality, [49] proposes two-stage optimization with 3D-GS for high-quality generation, [40] extends the single view Gaussian generation to sparse views for higher resolution. In this paper, we follow the feed-forward based image-to-3D generation scheme and propose a generalizable point-to-Gaussian model to advance the 3D-GS learning for image-to-3D generation.

## 3 METHOD

In this section, we first present the fundamental background of 3D Gaussian splatting (Section 3.1). Then, the Point to Gaussian generator, which takes the sparse point cloud generated from the pretrained 3d diffusion model and the paired images condition as inputs and outputs the 3D Gaussian representations, is introduced in (Section 3.2). To employ image conditions for further enhancing the geometric and texture features of 3D Gaussians, we present to integrate the 3D representations obtained from the **A**ttention, **P**rojection, and **P**oint feature extractor, termed **APP** Block in (Section 3.3) to enhance the cross modality features. Lastly, the loss function and data augmentation are introduced in (Section 3.4) for optimization.

### 3.1 Preliminary: 3D Gaussian Splatting

3D Gaussian Splatting (3D-GS) [14] shows high-fidelity rendering quality and real-time speed in novel view synthesis (NVS), which has gained a lot of popularity. 3D-GS renders images via splatting instead of volume rendering that is commonly used in implicit representation like NeRF [30]. Specifically, 3D-GS represents scenes with a set of explicit anisotropic 3D Gaussians. Each Gaussian distribution is defined by a 3D covariance matrix $\Sigma$ and a center position at point (mean) $\mathcal{X}$. A 3D Gaussian distribution $G(\mathcal{X})$ is formulated as follows:

$$G(\mathcal{X}) = e^{-\frac{1}{2}\mathcal{X}^T\Sigma^{-1}\mathcal{X}}. \tag{1}$$

For more effectively optimized by gradient decent, the 3D covariance matrix $\Sigma$ can be decomposed into a rotation matrix $\mathbf{R}$ and a scaling matrix $\mathbf{S}$:

$$\Sigma = \mathbf{R}\mathbf{S}\mathbf{S}^T\mathbf{R}^T, \tag{2}$$

where $\mathbf{R}$ and $\mathbf{S}$ are two learnable parameters. During optimization, the rotation matrix $R$ is transformed to quaternion $r$. Each Gaussian consists of an opacity $\sigma$ and spherical harmonics (SH) coefficients $c$ for rendering. Therefore, the complete Gaussian parameters are defined by $\mathcal{G} = \{(\mathcal{X}_i, \mathbf{S}_i, \mathbf{R}_i, \sigma_i, c_i)\}_{i=0}^n$. The Gaussians are projected from 3D space to 2D image plane for rasterization with viewing transform $\mathbf{W}$, then the 2D covariance matrix $\Sigma'$ can be computed as

$$\Sigma' = \mathbf{J}\mathbf{W}\Sigma\mathbf{W}^T\mathbf{J}^T, \tag{3}$$

where $\mathbf{J}$ is the Jacobian of the affine approximation of the projection transformation. Finally, the color $C$ of each pixel is accumulated by blending the overlapping Gaussians:

$$C = \Sigma_{i \in N}\alpha_i c_i \Pi_{j=1}^{i-1}(1 - \alpha_j), \tag{4}$$

where $\alpha_i$ is $\sigma_i$ multiplied by $\Sigma'$. The tile-based rasterizer is utilized for efficient forward and backward pass. In this paper, we reduce the degree of the SH coefficients in Gaussians to zero which represents only the diffuse color. We also remove the densification and pruning proposed in conventional per-scene 3D-GS optimization to adapt amortized optimization.

### 3.2 Point to Gaussian Generator

In this section, we introduce the architecture of our Point to Gaussian Generator. As shown in Fig. 3, the Point to Gaussian Generator shares the encoder-decoder structure, which converts the point

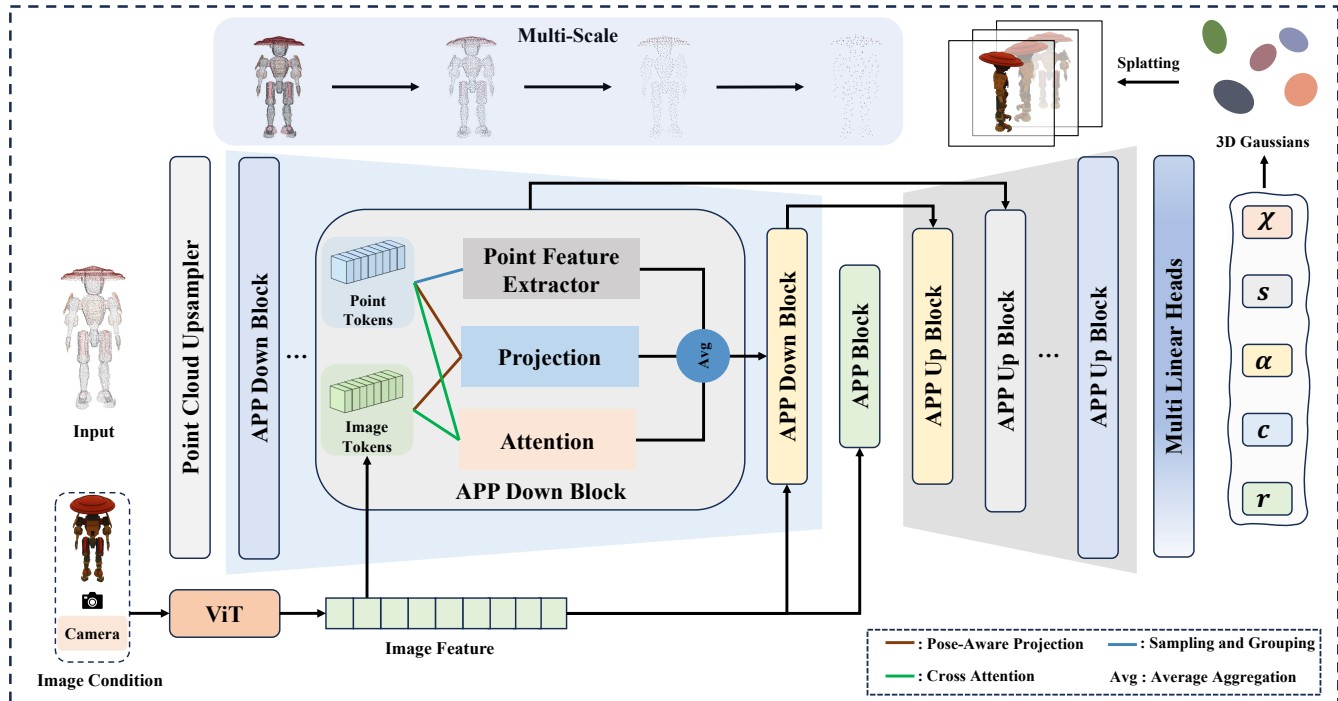

**Figure 3: Detailed architecture of Point-to-Gaussian Generator. Given the input point cloud and the corresponding image, a point cloud upsampler is first applied to increase the number of 3D points, followed by an encoder consisting of several APP Down Blocks to extract the multi-scale point cloud features. Each APP Block contains point feature extractor, projection and attention for cross modality feature enhancement. The point cloud features are decoded by a series of APP Up Blocks and Multi Linear Heads to obtain the final 3D Gaussians, and then the novel view images are obtained by conventional Gaussian splatting.**

cloud to 3D Gaussians. Specifically, we leverage the point cloud generated by a pretrained diffusion model [31] for initialization, and then upsample the points with densification operation. Meanwhile, the conditional images (could be single or multi) are also incorporated to enrich the Gaussian features. Finally, a multi-head gaussian decoder is incorporated to decode the features into Gaussian parameters for splatting. Denote that the point cloud with color is of $P_{N*6}$, conditioning images $\{I_v\}_{v=1}^{V}$ and camera parameters $\{C_v\}_{v=1}^{V}$, the output can be formulated as

$$\mathcal{G} = \Phi(P, \{I_v\}_{v=1}^{V}, \{C_v\}_{v=1}^{V}). \tag{5}$$

Here, the $\mathcal{G} = \{(\mathcal{X}_i, \mathbf{S}_i, \mathbf{R}_i, \sigma_i, c_i)\}_{i=0}^{N}$ represents the $N$ Gaussians.

*3.2.1 Point Cloud Upsampler.* To simplify the learning of 3D Gaussians, we utilize the point cloud as input. In single-scene optimization, as reported in the primitive 3D-GS [14], pruning and densification techniques are employed to adjust the Gaussian numbers.

Generally, sufficient number of 3D Gaussians can fairly represent the corresponding 3D objects, but an excess of Gaussians can also introduce computational and storage overheads. However, in the generalized 3D Gaussian framework, the gradient operates on the network, instead of the Gaussian itself, making it challenging to control and adjust the number of Gaussians dynamically. To strike a balance between performance and overhead, we initially perform a densification operation by upsampling the point cloud generated

from the pretrained 3D diffusion model, thereby augmenting the number of Gaussians in the network's final output. Specifically, we rely on the methodology outlined in [48] to implement a dense sampling operation on the point cloud.

*3.2.2 Multi-Scale Gaussian Decoder.* The Gaussian decoder's architecture adopts a U-Net structure, akin to that described in [23]. Following densification, as detailed in Section 3.2.1, the point cloud is inputted into the network. During downsampling, the number of point clouds progressively decreases, and the current layer's point cloud is derived via farthest point sampling (FPS) from the preceding shallower layer, thereby generating multi-scale point cloud features and expanding the receptive field. To further enrich the point cloud features and Gaussian attributes, we introduce projection and attention mechanisms for cross modality enhancement. For further elaboration, please refer to Section 3.3.

After obtaining the enhanced features, we introduce the multi linear heads, which utilizes multiple decoder heads for different attributes in the Gaussian. Since the coordinates of the point cloud and the Gaussian are not exactly the same in space, we take inspiration from [55] and learn the positional offset of the Gaussians concerning the point cloud, instead of the centers themselves, to simplify the learning process.

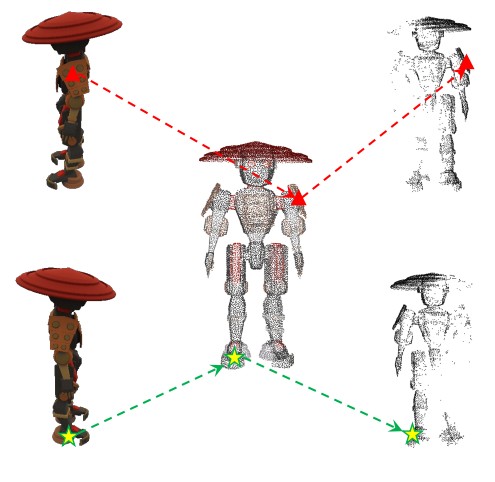

**Input Views**  **Point Cloud**  **Visible Points**

**Figure 4: Visualization of pose-aware projection. The red triangle and yellow star represents the corresponding position during projection. The self-occluded view only projects to the corresponding part of the point cloud during projection.**

## 3.3 Cross Modality Enhancement

In this section, we present the core component of the our Point to Gaussian generator, which integrate the 3D representations obtained from the **A**ttention mechanism, **P**rojection mechanism, and **P**oint feature extractor, termed as **APP**, for cross modality enhancement, as shown in Fig. 3. The point feature extractor we employ combines an efficient fusion of point-based methods [2] with the strong spatial inductive bias of voxel-based methods [56], which extracts the geometry and texture feature from the colored point cloud. Despite employing a multi-scale feature extraction module to provide a larger receptive field, the features extracted from point cloud input remain carries insufficient appearance feature and inaccurate geometry structural information to provide informative prior for precise GS generation. To further integrate the rich texture from the image modality into the point cloud tokens, we designed projection and attention modules which will be discussed in the following sections.

*3.3.1 Projection.* Numerous studies have been conducted in multimodal fusion, where an efficient and intuitive approach is projection. Inspired by this, we fuse multi-scale point cloud tokens with image tokens using projection, complementing each point cloud with a feature from the image modality based on its position in space and the view of the input image. Specifically, drawing on insights from [28], the projection considers the self-occlusion of the point cloud in space and employs a fast rasterization technique to map the pixel-wise image features to the visible points.

However, considering that the point cloud is assumed to be volumetric and non-transparent during the rasterization process, it is inevitable that points opposing the current camera viewpoint will be invisible during this process. As illustrated in Fig. 4, in the two provided viewpoints on the left side, we can only observe one side of the character, implying that the occluded point cloud on the other side will not participate in the projection process.

*3.3.2 Attention.* To further compensate for the point clouds that are against the current projection view, we propose using the attention mechanism to enhance the point cloud features further. Specifically, the point cloud features and image features interact via cross-attention, which further fuses the features of both modalities. It is worth noting that we do not explicitly define the mapping between the point cloud and the image but encourage the network itself to model the positional relationship between them. This approach differs from the projection process, where we align the point cloud and image based on the camera parameters and then explicitly fuse them.

This implicit fusion enables all points in the space to be augmented with image features obtained from DINOv2 [32]. Assuming the point cloud tokens are $T_p$, which are treated as the query, and the image tokens are $T_i$, which as key and value. The point cloud features interact with all the image tokens globally to obtain the output, which can be expressed as:

$$T_{att} = CrossAtt(T_p, T_i, T_i) \tag{6}$$

*3.3.3 Aggregation.* To enhance the features of the point cloud, we introduce projection and attention mechanisms for cross modality enhancement, respectively. Finally, we aggregate the enhanced features using average aggregation:

$$T_{out} = Avg(T_p, T_{pro}, T_{att}) \tag{7}$$

## 3.4 Optimization

*3.4.1 Training Loss.* The Gaussian generator is trained using rendering loss. In each iteration, we randomly select $M$ views, with one serving as the conditional input, while the remaining $M-1$ views are employed for supervision. Following the methodology of prior research [40], we compute the photometric loss and alpha loss between the rendered image and the ground truth image. Our objective is to minimize the following objective function.

$$L = L_{pixel} + \lambda_{pc}L_{pc} \tag{8}$$

$$L_{pc} = L_{cd} + L_{emd} \tag{9}$$

$$L_{pixel} = L_{MSE}^{rgb} + \lambda_{LPIPS}L_{LPIPS}^{rgb} + \lambda_\alpha L_{MSE}^\alpha \tag{10}$$

where $\lambda_{LPIPS}$ and $\lambda_\alpha$ are corresponding weight coefficient, and $L_{LPIPS}$ are perceptual image patch similarity [53].

*3.4.2 Data Augmentation.* During training, we use point cloud data from ground truth (GT), but when inference, the model inputs are generated from the 3D diffusion model. To mitigate the gap in data distributions, we perturb the point cloud data when training the model using data augmentation. Specifically, we jitter the coordinates and RGB values of the input point cloud by adding noise to them, thereby enhancing the robustness of the model to the input.

## 4 EXPERIMENTS

In this section, we initially delve into the specifics of the experiments, followed by a discussion on the dataset employed in our training and testing. Then we present our experimental results, offering both qualitative and quantitative analyses between ours

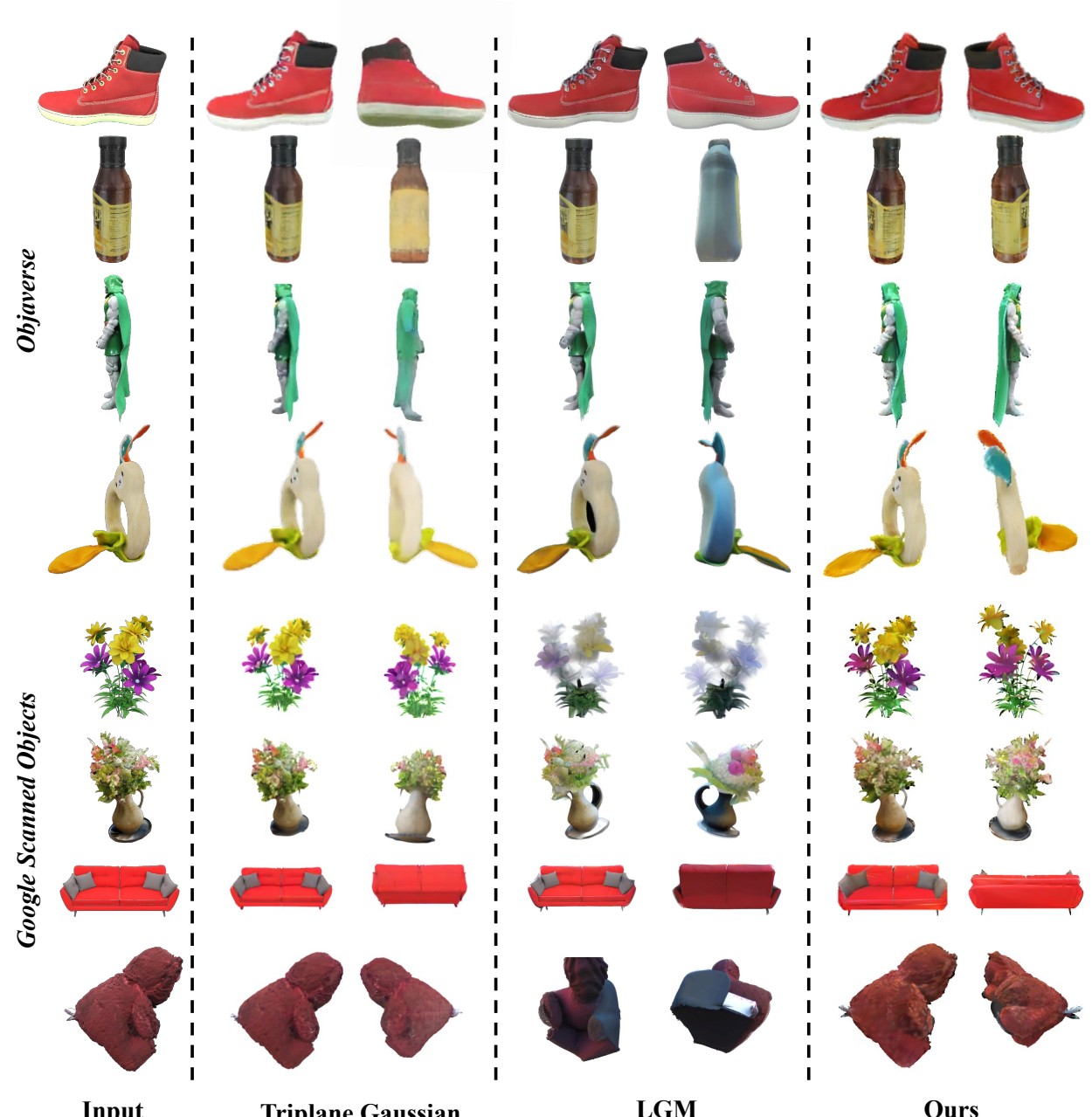

**Figure 5: Qualitative results among baselines of single view image-to-3D reconstruction on Objaverse [10] and Google Scanned Objects [11] dataset. Our method outperforms previous Gaussian based baselines in both geometry and texture representations, and in particular our method generates more consistent multi-view rendering results, thanks to the informative geometry prior of the point cloud input.**

and other methods. We conduct an ablation study to validate the effectiveness of our proposed modules lastly.

### 4.1 Implementation Details

The 3D diffusion model we deployed is Point-E [31], which presents commendable performance. Alternative methods for obtaining point clouds, such as DUSt3R [46], sparse reconstruction [25], or

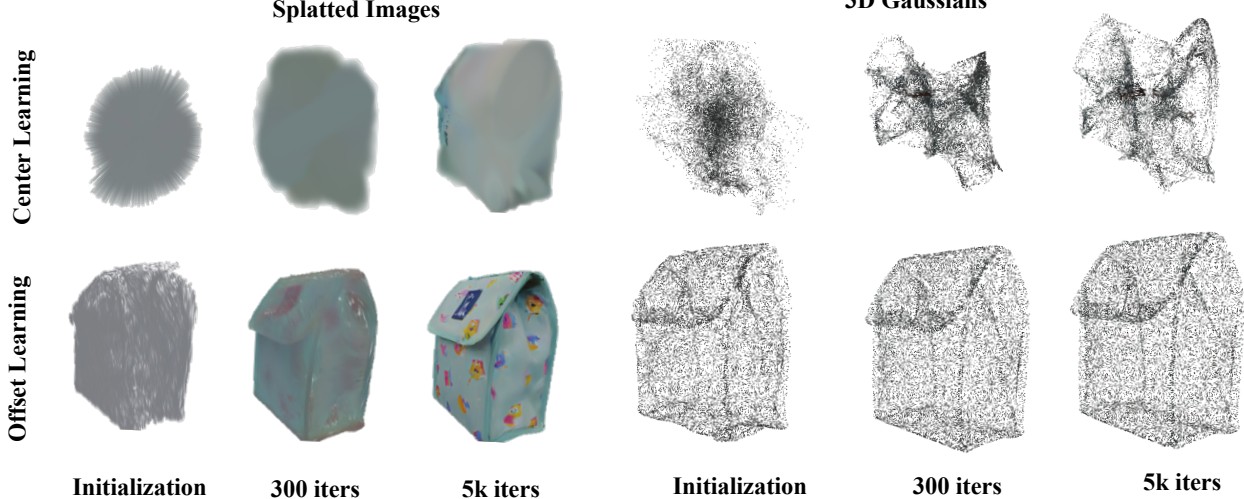

**Figure 6: Visualization of the training process. Three columns on the left represents splatted images from Gaussians in different iterations; Three columns on the right visualize the geometry of Gaussian centers during training. The first row represents that Gaussian centers are directly regressed while the second rows is that the learning of Gaussian centers' offsets.**

**Table 1: Qualitative rendering results on single-view image-to-3D. We compare the reconstruction quality for 32 novel views with 100 random objects selected from GSO [11]. Our method outperformed relevant feed-forward based generation baselines with comparable inference speed. The best results are bolded.**

|  | PSNR↑ | SSIM↑ | LPIPS↓ | Time↓ |
|---|---|---|---|---|
| One-2-3-45 [20] | 17.54 | 0.80 | 0.21 | ∼ 50s |
| Point-E [31] | 15.50 | 0.69 | 0.37 | ∼ 7s |
| LRM [12] | 16.09 | 0.79 | 0.28 | ∼ 6s |
| TriplaneGaussian [55] | 16.15 | **0.82** | 0.27 | ∼ **1s** |
| LGM [40] | 17.13 | 0.81 | 0.25 | ∼ 6s |
| Ours | **18.09** | **0.82** | **0.19** | ∼ 7s |

even 3D scanners, can also be utilized. The implementation of point cloud upsampler is modified form [48], which takes a 4K input and generates a 16K point cloud output. The image encoder utilized is the pretrained DINOv2 [32], and the architecture of our Point to Gaussian generator is based on the [23], with both the encoder and decoder consisting of four layers. Besides, to conserve computational resources, we incorporate the APP module solely into the encoder during the training process. The loss weights for mask and LPIPS losses are both set to 1. Meanwhile, the loss weights for point clouds are gradually attenuated from 1 to 0.05 using cosine annealing. During the training process, the number of views per iteration was set to 4 (i.e., $M = 4$). The experiment was conducted with 16 NVIDIA Tesla A100 GPUs for training, spanning approximately 3 days. The resolution of novel view rendering was 256, with a batch size of 128 (batch size 8 for each GPU).

## 4.2 Dataset

The model is trained on the Objaverse-LVIS [10] dataset. We render RGB images from 32 perspectives by rotating around the object's surface. Regarding the point cloud, we utilized the point cloud data provided by [26]. Through farthest point sampling, we obtained a 4k point cloud as the input. We evaluate our methods on the Objaverse and Google Scanned Objects [11] and randomly selected 100 test samples from each of the two datasets, respectively. For each case, akin to the training set, we rendered 32 different perspectives.

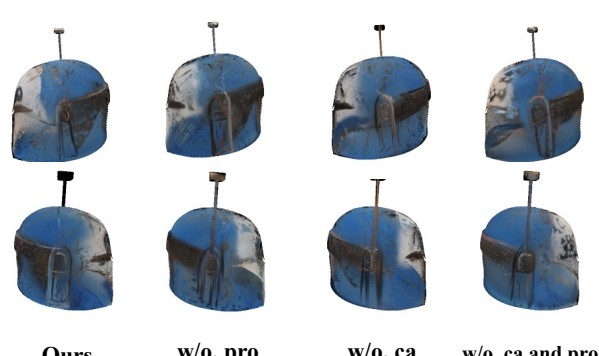

**Figure 7: Visualization of effects inside the proposed APP block. w/o.pro represents removing projection mechanism from APP block, and w/o. ca represents removing cross-attention mechanism, w/o. ca represents that the APP block contains only point feature extractor.**

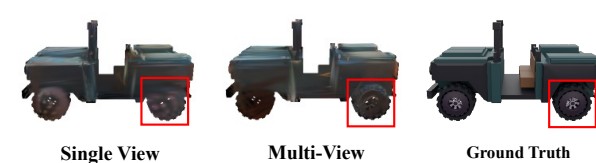

**Single View**  **Multi-View**  **Ground Truth**

**Figure 8: Visualization of rendered images and 3D Gaussians. the effect of different number of input views.**

### 4.3 Results

*4.3.1 Qualitative Comparison.* We predominantly compare our approach with recent Gaussian-based 3D generation methods on the Objaverse [10] and Google Scanned Objects [11]. The qualitative comparison of the results is depicted in Fig. 5. As observed, our method demonstrates richer high-frequency details compared to other models for both datasets. This outcome can be attributed to the cross modality enhancement, which incorporates projection and attention to optimally utilize the input image.

Reconstructing the backside view is a formidable challenge, as demonstrated by the results of LGM in the **Second** line. The results for the backside of the bottle appear inconsistent with the input view. However, our results exhibit remarkable consistency, and the Gaussians generated by our method more accurately recover the content from the input view, yielding more coherent and reasonable geometry and texture details. This improvement can be ascribed to the point cloud input, which simplifies the learning of Gaussians.

*4.3.2 Quantitative Comparison.* We also carried out a quantitative comparison with other methodologies and calculate PSNR, SSIM, and LPIPS [53] metrics to assess the quality of images. The quantitative results are presented in Table 1. As evident from the table, our method surpasses other feed-forward methods in terms of generation quality across all metrics, while maintaining a comparable inference speed. The time intensity of our process is primarily attributed to the initial stage of point cloud acquisition, as can be deduced by comparing our time usage with that of Point-E. However, in the second stage, the Point to Gaussian Generator can be inferred in real-time due to the fast rendering speed of splatting.

### 4.4 Ablation Study

*4.4.1 APP Block.* We first trained a baseline model using the point feature extractor only. Based on it, we added projection and attention respectively. The quantitative experimental results are presented in Table 2, which indicates the projection and attention we designed can significantly facilitate the learning of Gaussian attributes. Moreover, when both mechanism are employed, which further enhances the performance. A qualitative comparison of the results, is shown in Fig. 7, which also demonstrates that the quality of the reconstructed images is significantly improved.

*4.4.2 Number of Views.* With the assistance of existing image diffusion models, for example, MVDream [38], our method can also support multi-image input. To be specific, we first convert the single-image input into four consistent images and then feed them into

**Table 2: Ablation study on our proposed APP block. The best results are bolded.**

| Point Feature Extractor | Attention | Projection | PSNR↑ | SSIM↑ | LPIPS↓ |
|---|---|---|---|---|---|
| ✔ | | | 15.84 | 0.72 | 0.31 |
| ✔ | | ✔ | 17.17 | 0.78 | 0.22 |
| ✔ | ✔ | | 17.25 | 0.80 | 0.25 |
| ✔ | ✔ | ✔ | **17.92** | **0.81** | **0.21** |

**Table 3: Ablation study on the number of input. Given multi-view image input, our proposed Point-to-Gaussian Generator achieves better reconstruction results compared to single-view image input.**

| | Single View | Multi View |
|---|---|---|
| PSNR↑ | 17.92 | 18.09 |
| SSIM↑ | 0.81 | 0.82 |
| LPIPS↓ | 0.21 | 0.19 |

the Gaussian generator. The results are presented in Section 4.4.2 indicating that more image input can further improve the reconstruction quality compared to single. The qualitative results in Fig. 8 also support this conclusion, as the reconstructed image with multi-image input exhibits better clarity and geometric consistency.

*4.4.3 Learning Offsets for Gaussian Centers.* To demonstrate the efficiency of Gaussian center offsets learning, we conducted a comparative analysis between involving Gaussian center offsets learning and direct Gaussian centers learning. We visualize the rendered images and 3D Gaussians at different iterations, as displayed in Fig. 6. The top row presents the outcome of direct Gaussian centers learning. At the beginning of training, the 3D Gaussians are predominantly characterized by noise, and the model requires approximately **5k** iterations before revealing the basic shape of the bag. In contrast, with offset learning, as depicted in the bottom row, the 3D Gaussians exhibit good geometry during the initial training and demonstrate faster convergence speed. After just **300** iterations, the model can reconstruct the basic geometry of the input object. With offset learning, the Gaussians can be initialized to a suitable position by the point cloud at the beginning of training. This approach allows the model to concentrate more on modeling other Gaussian attributes and is considerably simpler than mapping from images to Gaussians.

### 5 CONCLUSION

In this paper, we present a large point-to-Gaussian model for image-to-3D generation. Point-to-Gaussian model inputs point cloud to generate Gaussian parameters, in which the input point cloud is generated from a large 3D diffusion model (e.g. Point-E) from the 2D image. With the geometry prior from point cloud, Point-to-Gaussian model is able to significantly improve image-to-3D generation. In addition, based on a multi-scale network, we further devise a APP block, which fuse the image features into point cloud representations with an attention mechanism and pose-aware projection mechanism. Extensive experiments demonstrate the effectiveness of the proposed approach on GSO and Objaverse datasets, and the proposed method achieves state-of-the-art performance.

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
