# OpenReview forum: "Large Point-to-Gaussian Model for Image-to-3D Generation"
_acmmm.org/ACMMM/2024/Conference — MM2024 Poster_

### Official Review · Reviewer_bqhu · 2024-04-30

**Rating:** 3
**Confidence:** 3

**Summary:**

- Task: single img to 3DGS
- Contribution:
    - not 'img -> 3DGS', but 'img-> point cloud -> 3DGS'
    - propose APP block for fusing image and pc feature
    - SOTA performance, even trained with a much smaller dataset (objaverse-LVIS)

- Method：
    - Get initial pc from input img by PointE
    - Upsample the pc by SnowFlakeNet
    - Send the pc and img to a designed network called Point-to-Gaussian Generator, to get 3DGS
        - Point-to-Gaussian Generator mainly consists of some APP blocks. The final output heads output 3D Gaussian parameters。
        - An APP block contains point feature extractor, projection（project img features to 3D points？）and cross attention. The final result is the average of the three.

Summary:
This paper is well-written. The two main contributions are clearly stated, reasonable, and sound novel to me. My main concern is that the qualitative results in Figure 6 do not seem convincing enough. Some input images are from weird perspectives. Lack qualitative comparisons with the baseline method with the highest quantitative results (one-2-3-45). and sometimes the improvement compared to baseline methods does not seem significant. If the authors can provide more convincing qualitative results, and explain in detail on the evaluation strategy  (see the limitations part), I may be convinced. Also, there are some suggestions to make the 'Method' section clearer and easier to understand. Overall, I tend to regard this paper as 'borderline'. If the above concerns can be addressed, especially more convincing visual comparisons with baseline methods especially one-2-3-45, I may tend to accept it.

**Strengths:**

The novelty sounds sufficient to me, with the two main contributions:

1) introducing 3D point cloud as a middle step instead of directly generating 3DGS from imgs, and

2)  proposing a APP block for fusing image and point features.

The motivation is clearly stated and convincing.

Specially, the introduction and related work are well-written and easy to follow.


The overall language is good.

**Limitations:**

The main limitations lie in the experiment and method sections. Besides, some other small suggestions are provided.
## Experiment
**Experimental results not convincing enough.**
Among all 5 baselines (one-2-3-45, Point-E, LRM, TriplaneGaussian, and LGM) listed in Table 1, one-2-3-45 achieves the overall highest scores. However, some other listed baselines are actually more recent and they claim to be better than one-2-3-45. For example， the paper of TriplaneGaussian shows that it outperforms one-2-3-45 by a significant margin, both quantitative and qualitative. Notably, for evaluation, TriplaneGaussian also randomly selected 100 shapes from the Google Scanned Objects dataset, just like this paper. That is to say, the conclusion of TriplaneGaussian and this paper kind of disagree with each other. However, the visual comparisons in TriplaneGaussian seem more convincing than this paper, showing a more significant improvement of TriplaneGaussian compared to baseline methods. Although Figure 5 of this paper shows that the proposed method outperforms baseline methods to some degree, the superiority is not as significant. Another example would be LGM, which claims to outperform DreamGaussian, and DreamGaussian claims to outperform one-2-3-45.

It would be much more convincing if more qualitative comparisons could be provided, with a significant margin compared to baseline methods. However, the supplementary file only provides visual results of the proposed method, lacking comparison with baseline methods.


**Evaluation Strategy.** During evaluation, how did the authors choose the viewpoint of input single image? In line 787, it's said that 'For each case, akin to the training set, we rendered 32 different perspectives.' Does it mean that for each shape, 32 viewpoints are used as input to generate 3DGS respectively followed by evaluation, or only one viewpoint is used as input to generate 3DGS, but 32 viewpoints are used to render the generated 3D result for evaluation?

In Figure 5, some input images are from a side viewpoint, like rows 2,3, and 4. Under such side-view input circumstances, the proposed method shows a better performance, especially regarding the opposite side of the input. Is it that, the proposed method has an advantage for such cases with a not very ideal input viewpoint? When the input image is from the front or maybe 2 o'clock direction, does it still show an advantage?

Considering the application of reconstructing a 3D model from a single image, one is likely to take a photo of the target from the front or maybe 2 o'clock direction, so that the target can be clearly seen in the image. Using a side-view image as input is indeed more challenging, but does not seem as necessary, since it's often not any harder to take a photo from the front.


**About the multi-view experiment.** The proposed method is an img-to-3DGS method. MVDream is an img-to-NerF method. They focus on a quite similar task. But section 4.4.2 proposes to first use MVDream to generate 4 novel view images, then use all images as input of the proposed method. Why don't we directly use MVDream, why bother using the proposed method? Compared to directly using MVDream (further optimizing a NerF to fit the generated multi-view images), does the proposed method show a better performance?

**Some other suggestions about experiments**
- In Figure 7 (ablation study of the proposed APP block)， it's suggested to add GT, since the presented object is a little abstract and the readers don't really know what it's supposed to look like. If GT was added, it would be clearer for the readers to see how the APP module outperforms other settings.


- Add GT in Figure 5 (main qualitative comparison results)
- In Figure 5 (main qualitative comparison results), Why are there only 3DGS-based baselines? one-2-3-45 seems stronger according to Table 1, so why not compare with it qualitatively?


## Method
The proposed method can be presented more clearly. See the following:
- it's suggested to add mathematical notations of the input and output of APP Block, especially the dimensions of point tokens and image tokens.
- **How are the multi-scale pcs used.** The multi-scale pc obtained from FPS, where are they sent to the network? In Figure 3, there are no lines to connect them to the network. Are they only sent to the point extractor for multi-scale feature extraction, or is the whole network repeated several times with different scales of pc as input?
- **Details about Point Feature Extractor.** It's only said in line 496 that the point feature extractor combines point-based methods [2] and voxel-based methods [56], but how? It's ok to refer the readers to the supplementary file, instead of not saying anything in the main paper.
- **Implementation of projection.**
    - It's said in line 513 that 'drawing on insights from [28], the projection ....'. This sounds amphibolous. Is it exactly the same as [28]? If so, please just say 'we directly adopt the method in [28]'. Otherwise, please write clearly what modifications are there compared to [28].
    - Besides, for readers not familiar with [28], the so-called 'self-occluded' can be a little confusing. We can see in Figure 4 that, if look from one side, the opposite side is with fewer points due to occlusion. But shouldn't it be no points at all on the opposite side since it's completely occluded and no points should be seen?
- **How 'projection' fuses the img and point features.** I see that 'projection' maps img pixels with 3D points. But what then? How are the img and point features fused? Does it cat the img features to the corresponding points? or maybe sum?
- **How are the point tokens obtained?** Figure 3 shows that image tokens are obtained from image features of ViT, but how about the point tokens?
- **Details about the 32 perspectives.** In line 781, it's said that ''We render RGB images from 32 perspectives by rotating around the object's surface''. How to rotate in detail? It's ok to refer the readers to the supplementary file or say 'we follow XXX'


Some other small questions or suggestions:

- Data Augmentation： During training, why use perturbed GT as input, instead of using results of diffusion models like in inference? If it's because the diffusion model takes too long time, it would be better to add something like 'considering diffusion model's XXX'.
- In line 745, why is it OK to only incorporate the APP module into the encoder during training?
- In Section 4.4.3, it's said that directly learning Gaussian centers instead of offsets would take about 5k iterations to show an overall shape, while learning offsets only takes 300. It's suggested to also tell the readers how many iterations in total would there be.


## Some details or suggestions regarding language or presentation
- In line 536: ''This implicit fusion enables all points in the space to be augmented with image features obtained from DINOv2 [32]''. It's a bit weird to suddenly talk about the image encoder in the Attention part. The image features are actually already used in the previous step 'projection', right?
- When citing SnowFlakeNet, like in Line 740 or 441, it's suggested to cite it with its name instead of only a number. i.e. say something like ''SnowFlakeNet [48]'' instead of just '[48]'. As a module used in the proposed method, it would be more convenient for the readers who are familiar with this method, and they no longer need to click to see what is [48].
- It may be more clear for the readers, if the number of samples of each used dataset is presented when introducing the datasets.(objaverse-LVIS,GSO, 和objaverse)

- Line 501: ''remain carries insufficient appearance feature'', 'remain carries' seems incorrect regarding grammer
- Line 507: ''following sections'' -> ''following subsections''?
- Line 890: ''The results are presented in Section 4.4.2'', （it is Section 4.4.2 already. Do you mean Table 3 instead?）

**Suitability:**

3

---

### Official Review · Reviewer_fDsM · 2024-05-01

**Rating:** 5
**Confidence:** 2

**Summary:**

This paper proposes a novel generative model that transforms large point clouds into Gaussian representations. According to the complete pipeline diagram, the model takes a point cloud generated by a pre-trained 3D diffusion model and a two-dimensional image as input.   It employs the APP block (Attention mechanism, Projection mechanism, and Point feature extractor), as proposed by the author, to integrate image and point cloud features.   Ultimately, a 3D Gaussian image is produced using these integrated features, which is then converted into a multi-view image through traditional Gaussian scattering.

**Strengths:**

1.The use of cross-modal information to enhance traditional Gaussian point cloud generation is highly innovative.
2.The architecture diagram of the Point-to-Gaussian generator is detailed and clear.
3.The experimental section demonstrates superior performance compared to LRM, Triplane Gaussian, and LGM. Results on the Objaverse and Google Scanned Objects datasets reveal richer high-frequency details.

**Limitations:**

1.The paper lacks a detailed implementation description of the APP Up block operations following the Average Aggregation in Figure 3.
2.As the author mentioned in Section 4.3.1, reconstructing the back view is a difficult challenge, which the author can further think about how to improve the optimization.
3.This paper lacks a detailed description of the full English names of PSNR, SSIM, and LPIPS metrics in Section 4.3.2.

**Suitability:**

3

---

### Official Review · Reviewer_MqPM · 2024-05-20

**Rating:** 2
**Confidence:** 3

**Summary:**

This work introduces a large Point-to-Gaussian model to generate high-quality 3D Gaussians with point cloud input. Firstly, the initial point cloud is produced from large 3D diffusion model conditional on 2D image to generate the Gaussian parameters, which provides initial 3D geometry prior for Gaussian generation, thus significantly facilitating image-to- 3D Generation. Moreover, the Attention mechanism, Projection mechanism, and Point feature extractor, dubbed as APP block, is proposed for fusing the image features with point cloud features. The proposed method was evaluated on GSO and Objaverse datasets for image-to-3D Generation and compared against recent works.

**Strengths:**

1.Clarity and readability: the paper is well written, and the language is easy to follow.
2.The proposed method is the first attempt to utilize the generalizable Point-to-Gaussian Generator for feed-forward image-to-3D generation. The Attention mechanism, Projection mechanism, and Point feature extractor as APP block was designed into Point-to-Gaussian generator for Cross Modality Enhancement. The proposed method is somewhat innovative in overall scheme design, but the core module has weak innovative.
3.Comparison against recent works.

**Limitations:**

1. As noted in Sec3.2, the point cloud was generated by a pretrained diffusion model [31] for initialization. So, what if the initial point cloud generated by the diffusion model produces suboptimal results?
2. The Point Cloud Upsampler of Point-to-Gaussian Generator utilizes the methodology outlined in [48] to implement a dense sampling operation on the point cloud. However, the proposed method in [48] is aimed at point cloud completion. How to use it to generate dense point clouds? The technical details are not clearly explained in the paper. At the same time, the function of this module was not verified in the ablation experiment.

**Suitability:**

3

---

### Official Review · Reviewer_jmMt · 2024-05-24

**Rating:** 2
**Confidence:** 4

**Summary:**

The authors propose a new approach: a large Point-to-Gaussian model. This model takes an initial point cloud (created from a 2D image) and uses it to generate the 3D model. The point cloud provides valuable 3D information that aids the generation process. Additionally, the paper introduces an APP block that merges image and point cloud features for improved results. Experiments show that this method achieves superior performance on standard datasets.

**Strengths:**

1. Their method is the first attempt to utilize the generalizable Point-to-Gaussian Generator for feed-forward image-to-3D generation.
2. They introduce APP block into Point-to-Gaussian generator for Cross Modality Enhancement, which further integrate the geometric structural features with the 2D texture features for more effective learning.

**Limitations:**

1. The experimental results are incorrect. Table 1 shows the results of **single-view** image-to-3D, but the results in Table 1 are the same as the results for **Multi View** in Table 3 Ablation study on the number of input. Therefore, Table 1 does not compare the results of single view.
2. The experimental results are insufficient. For the image-to-3D setting, the experimental section only compares the results of novel view synthesis, and does not compare reconstruction results, such as CD, IoU metrics that were compared in the baseline.
3. I have doubts about the final results of the paper. To my knowledge, the point clouds generated by Point-E from a single image are far inferior to the ground truth (GT) point clouds. The authors claim to have performed data augmentation on the GT point clouds during training: "we jitter the coordinates and RGB values of the input point cloud by adding noise to them, thereby enhancing the robustness of the model to the input." However, even such simple perturbations still result in significantly better results than the point clouds generated by Point-E, especially in invisible views.
4. LRM has a much larger training dataset than the one used in this paper. LGM builds upon MVDream, which utilizes 2D priors and more 3D data. In contrast, the proposed method is trained solely on the Objaverse-Lvis dataset. Given that Point-E struggles to accurately reconstruct the backside of input images, I am skeptical about the ability of the proposed method to generate accurate reconstructions from such point clouds, especially for the backside. The APP structure employed in the paper is a common approach. Therefore, I would like the authors to provide more convincing evidence to support their claim that the proposed method outperforms LGM and LRM in reconstructing the backside.
5. Here are some small writing errors I found in the paper: Line 890: "The results are presented in Section 4.4.2" should be "The results are presented in Table 3".

**Suitability:**

3

---

### Meta-Review · Area_Chair_RaZ2 · 2024-06-29

**Recommendation:** Accept (Poster)
**Confidence:** 4

**Metareview:**

Pros: The paper is easy to follow, and the idea is relatively new.
Cons: There are some errors regarding the empirical results reported in Tables. As the topic is very hot, there have been many recent efforts, including e.g. TriplaneGaussian and many others, the authors are encouraged to follow closely and report comparisons with them, or at least the major ones. There is some concern of lack of details in method presentation, e.g. when describing the APP Up block operations, for readers to replicate the work. Moreover, there is some doubt of the empirical performance of the work.